# Incidence and risk factors for post-stroke delirium in the elderly: A national inpatient sample (NIS) analysis

Jianrong Zhang[1☺], Mei Chen[2☺], Yaoyang Huo[3☺], Xilin Liu[2☺], Yu'e Wu[4], Xiaohuan Li[2], Jingqin Wang[4], Fengling Yang[5], Gang Liu[5], Hao Xie[6*], Ying Gao[7*]

1 Department of Nursing, Houjie Clinical College of Guangdong Medical University, Houjie Hospital, Dongguan, Guangdong, China, 2 School of Nursing, Guangdong Medical University, Dongguan, Guangdong, China, 3 The Second School of Clinical Medicine, Southern Medical University, Guangzhou, Guangdong, China, 4 Department of Neurology, Houjie Clinical College of Guangdong Medical University, Houjie Hospital, Dongguan, Guangdong, China, 5 Department of Critical Care Medicine, Houjie Clinical College of Guangdong Medical University, Houjie Hospital, Dongguan, Guangdong, China, 6 Division of Orthopaedic Surgery, Department of Orthopaedics, Nanfang Hospital, Southern Medical University, Guangzhou, Guangdong, China, 7 Department of Gastrointestinal Infection, Houjie Clinical College of Guangdong Medical University, Houjie Hospital, Dongguan, Guangdong, China

☺ These authors contributed equally to this work.
* 13711837955@163.com (YG); hao_xie2018@163.com (HX)

## Abstract

### Background

Post-stroke delirium (PSD) is a critical neuropsychiatric condition affecting up to 50% of elderly patients during hospitalization, often leading to poorer outcomes. Despite its prevalence, PSD remains underrecognized in clinical practice, and national-level studies exploring its risk factors are limited.

### Objective

To examine the incidence and risk factors associated with PSD in elderly individuals (≥65 years) using a large, nationally representative dataset.

### Methods

Data from the Healthcare Cost and Utilization Project National Inpatient Sample (2010–2019) were analyzed. Elderly patients with a primary diagnosis of stroke were selected, and PSD was defined using the International Classification of Diseases, Ninth Revision, Clinical Modification (ICD-9-CM) and ICD-10-CM codes. Multivariate logistic regression, adjusted for demographic, clinical, and hospital variables, identified independent PSD risk factors.

**Data availability statement:** The data underlying the results presented in the study are available from https://www.hcup-us.ahrq.gov/db/nation/nis/nisdbdocumentation.jsp.

**Funding:** "Dongguan Science and Technology Bureau (Grant number: 20231800936172), MD Ying Gao.".

**Competing interests:** The authors have declared that no competing interests exist.

**Abbreviations:** PSD, Post-stroke delirium; HCUP, Healthcare Cost and Utilization Project; NIS, National Inpatient Sample; AHRQ, Agency for Healthcare Research and Quality; LOS, Length of stay; AIDS, Acquired immunodeficiency syndrome; ICU, Intensive-Care-Unit; OR, Odds ratios; CI, Confidence intervals; ICD-9-CM, International Classifcation of Diseases (Ninth Edition) Clinical Modifcation; ICD-10-CM, International Classifcation of Diseases (Tenth Edition) Clinical Modifcation.

## Results

Among 1,644,773 elderly stroke patients, the incidence of PSD was 19.5%. PSD occurred in 18.9% of ischemic strokes and 24.7% of hemorrhagic strokes. Patients with PSD were significantly older, with a median age of 79 years, compared to 78 years in those without PSD ($p < 0.001$). They also experienced prolonged hospital stays (5 days vs. 4 days, $p < 0.001$), incurred greater hospitalization costs ($44,863 vs. $35,787, $p < 0.001$), and exhibited a higher risk of in-hospital mortality (12.6% vs. 7.0%, $p < 0.001$). Major risk factors for PSD include: sepsis (OR = 2.364, 95%CI = 2.329–2.400), three or more comorbidities (OR = 2.049, 95%CI = 1.984–2.116) and fluid/electrolyte disorders (OR = 1.902, 95%CI = 1.886–1.918), psychoses (OR = 1.765, 95%CI = 1.725–1.806).

## Conclusions

PSD is frequently observed in elderly stroke patients and is associated with adverse clinical outcomes. Advanced age, comorbidities, and stroke-related complications are significant risk factors. These results underscore the importance of developing focused prevention and intervention strategies to enhance outcomes for this high-risk population.

## Introduction

Delirium is a severe acute neuropsychiatric syndrome [1], representing a loss of brain function in response to pathophysiological stress and occurring commonly following acute stroke [2]. Previous studies have documented that the prevalence of post-stroke delirium varies widely, ranging from 20% to 35% [2–4]. In addition, a systematic review indicated a clear trend where older age correlates with an increased risk of developing delirium post-stroke, with prevalence rates of 20% for ages 60–64, 25% for 65–74, and 34% for those aged 75–79 [3]. Post-stroke delirium manifests as sudden cognitive disruptions that impact attention, memory, language, and visuospatial processing [5]. Specifically, patients exhibit a reduced capacity to concentrate, sustain, or shift their attention [6]. Post-stroke delirium is closely linked to adverse outcomes. Studies have shown that post-stroke delirium is associated with higher in-hospital mortality (12.3% vs. 7.8%) and a substantial medical cost burden, contributing to healthcare costs exceeding $164 billion annually [2,7]. Furthermore, it is associated with prolonged hospital stays, worsened functional and cognitive impairment, greater rehabilitation needs, and a higher likelihood of transfer to ICU (Intensive-Care-Unit) inpatient care [2,3,5]. Consequently, post-stroke delirium poses a significant challenge in stroke management and rehabilitation.

The occurrence of post-stroke delirium is influenced by a complex interplay of risk factors, which can be classified into susceptibility and predisposing factors. Susceptibility factors include advanced age, male gender, pre-stroke cognitive impairment, and various systemic or metabolic diseases [8]. Predisposing factors reported in the

literature include a high comorbidity burden, depression, and a history of alcohol abuse [9,10]. Despite the high susceptibility of stroke patients to delirium, current clinical guidelines often fail to prioritize delirium management in stroke care [5]. This oversight results in inadequate detection and management strategies, emphasizing the need for heightened awareness among healthcare professionals [11]. Addressing post-stroke delirium is therefore crucial.

Prior studies investigating post-stroke delirium have predominantly been single-center studies, often focusing on specific geographic regions or patient subgroups, creating a gap in understanding national patterns and demographic variations in the risk of post-stroke delirium [12]. This study sought to address this gap by utilizing a large, nationally representative database to explore the incidence and risk factors for delirium in older adults with stroke. The findings from this study are intended to attract widespread attention from clinical experts and provide a robust reference for the prospective management of delirium in this population.

## Materials and methods

### Data source

Data for this study were obtained from the Healthcare Cost and Utilization Project (HCUP) National Inpatient Sample (NIS), the largest publicly available all-payer inpatient healthcare database in the United States. Funded by the Agency for Healthcare Research and Quality (AHRQ), the NIS employs a stratified sample drawn from over 1,000 hospitals across 46 states, representing approximately 20% of all U.S. hospital admissions each year [13]. The database contains comprehensive information, including patient demographics, admission status, diagnoses, procedures, comorbidities, hospital characteristics, insurance type, length of stay (LOS), in-hospital mortality, total charges, and discharge disposition. All classifications were encoded according to the International Classification of Diseases, Ninth Revision, Clinical Modification (ICD-9-CM) and ICD-10-CM standards. As this study utilized anonymized, publicly available data, it was not subject to ethical approval protocols.

### Data collection

Elderly patients (aged ≥65 years) hospitalized between January 1, 2010, and December 31, 2019, with a primary diagnosis of stroke (identified using ICD-9-CM and ICD-10-CM codes) were included in this retrospective cohort study. Ischemic stroke, hemorrhagic stroke and post-stroke delirium were identified using ICD-9-CM and ICD-10-CM diagnostic codes (S1 File). Patients with incomplete data, those under 65 years of age, and those admitted for non-stroke diagnoses were excluded. The final study cohort was stratified into two groups based on the presence or absence of delirium following stroke. Demographic data (age, race, sex), clinical characteristics (length of stay [LOS], hospitalization costs, admission mode), medical complications (e.g., pneumonia, sepsis, acute renal failure, urinary tract infections, acute myocardial infarction), and pre-existing comorbidities (e.g., fluid and electrolyte disorders, psychoses) were collected (Table 1). Fig 1 delineates both the exclusion criteria and the data collection process.

### Data analysis

Statistical analyses were performed with SPSS version 25 (IBM SPSS Statistics, USA). The normality of continuous variables was evaluated using the Kolmogorov-Smirnov test. We applied non-parametric Wilcoxon rank-sum tests for continuous variables and chi-square or Fisher's exact tests for categorical variables. Univariate and multivariate logistic regression analyses were performed to identify independent risk factors for post-stroke delirium; before fitting the multivariate models, multicollinearity diagnostics were conducted separately for the two predictor categories (comorbidities and complications) [14]. In addition, sensitivity analyses were stratified by stroke subtype, and risk factors for delirium were examined separately for ischemic and hemorrhagic strokes. These analyses adjusted for demographic factors, comorbidities, hospital characteristics, and complications. Odds ratios (OR) with 95% confidence intervals (CI) were computed. Considering the extensive sample sizes in previous NIS studies, statistical significance was defined as $p \leq 0.001$ [13].

**Table 1. Variables entered into the binary logistic regression analysis.**

| Variables Categories | Specific Variables |
|---|---|
| Patient demographics | Age (65–79, ≥ 80 years), sex (male and female), race (White, Black, Hispanic, Asian or Pacific Islander, Native American and Other) |
| Hospital characteristics | Type of admission (non-elective, elective), bed size of hospital (small, medium, large), teaching status of hospital (nonteaching, teaching), location of hospital (rural, urban), type of insurance (medicare, medicaid, private insurance, self-pay, no charge, other), location of the hospital (northeast, midwest or north central, south, west) |
| Comorbidities | AIDS, alcohol abuse, deficiency anemia, chronic blood loss anemia, congestive heart failure, coagulopathy, depression, diabetes (with chronic complications), drug abuse, hypothyroidism, liver disease, fluid and electrolyte disorders, psychoses, pulmonary circulation disorders, renal failure, peptic ulcer disease excluding bleeding, weight loss, rheumatoid arthritis/collagen vascular diseases |
| Complications | Dysphagia, acute myocardial infarction, pneumonia, urinary tract infection, deep vein thrombosis, pulmonary embolism, sepsis |

AIDS: Acquired immunodeficiency syndrome.

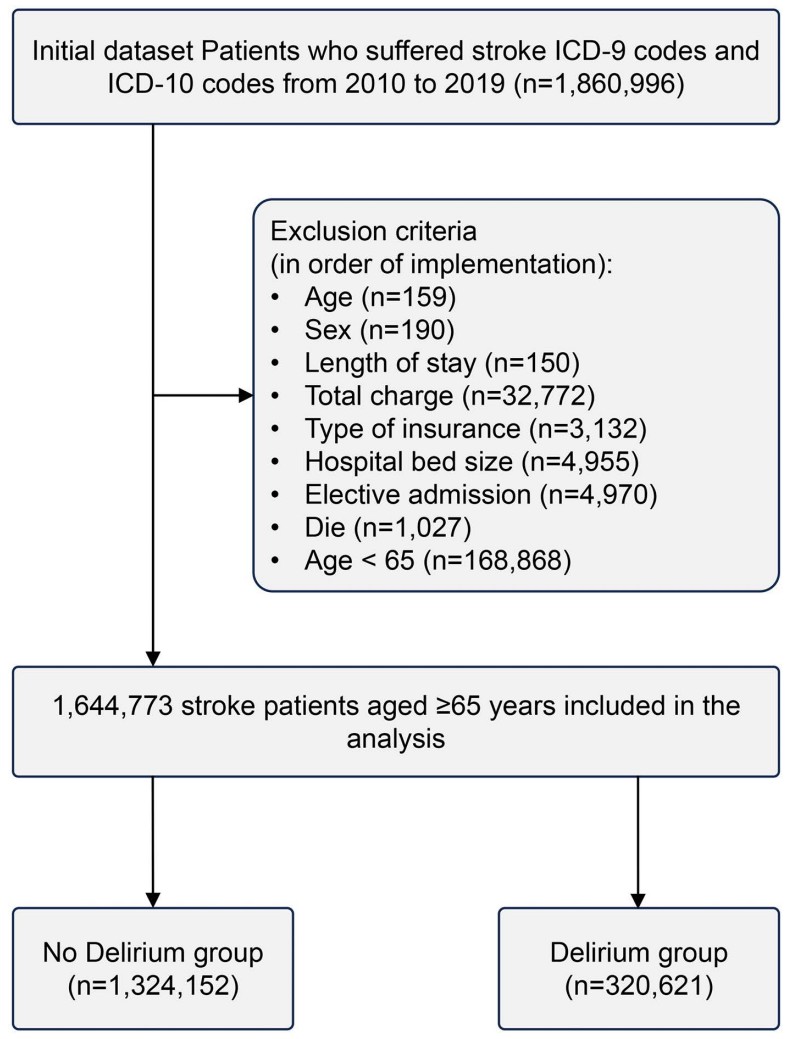

**Fig 1. Flowchart depicting the study population selection process.**

## Results

### Characteristics related to demographics and hospital factors for patients in both groups

Between 2010 and 2019, the NIS database recorded 1,644,773 elderly stroke patients. Of these, 320,621 cases of PSD were documented, corresponding to an incidence rate of 19.5% (Table 2). PSD occurred in 18.9% of ischemic strokes and 24.7% of hemorrhagic strokes (S1 Table). Moreover, patients who developed delirium were significantly older, with a median age of 79 years, compared to 78 years among those without delirium ($p<0.001$). The delirium group had a greater proportion of patients aged ≥80 years (43.5% vs. 43.2%, $p<0.001$). A significant difference in gender distribution was observed between the two groups ($p=0.001$). Regarding race, a higher proportion of Black patients (11.7% vs. 10.4%, $p<0.001$) and Asian/Pacific Islander patients (2.7% vs. 2.5%, $p<0.001$) developed delirium compared to the non-delirium group. Patients in the delirium group exhibited a significantly extended median hospital stay (5 days vs. 4 days, $p<0.001$) and incurred greater total hospital costs ($44,863 vs. $35,787, $p<0.001$). Furthermore, patients with delirium experienced a higher in-hospital mortality rate (12.6% vs. 7.0%, $p<0.001$) (Table 2).

### Risk factors associated with delirium after stroke in the elderly

Collinearity was assessed using variance inflation factors; all independent variables returned VIF values below 5, indicating that multicollinearity was not a concern (S5 Table) [14]. Multivariate logistic regression analysis identified several significant independent risk factors for post-stroke delirium (Tables 3 and 4): age ≥80 years (OR=1.237; 95% CI== 1.227–1.247, $p<0.001$), Black race (OR = 1.113; 95% CI = 1.099–1.127), Asian or Pacific Islander race (OR = 1.060; 95% CI = 1.034–1.087), Other race (OR = 1.094; 95% CI = 1.077–1.110), ≥3 comorbidities (OR = 2.049; 95% CI = 1.984–2.116), large hospital bed size (OR = 1.083; 95% CI = 1.070–1.095), elective admission (OR = 0.681; 95% CI = 0.672–0.690), geographic region: Midwest/North Central (OR = 1.200; 95% CI = 1.185–1.215), South (OR = 1.250; 95% CI = 1.235–1.264), and West (OR = 1.290; 95% CI = 1.272–1.307), alcohol abuse (OR = 1.126; 95% CI = 1.099–1.153), deficiency anemia (OR = 1.150; 95% CI = 1.136–1.164), congestive heart failure (OR = 1.186; 95% CI = 1.174–1.198), coagulopathy (OR = 1.333; 95% CI = 1.314–1.353), depression (OR = 1.236; 95% CI = 1.221–1.251), diabetes with chronic complications (OR = 1.062; 95% CI = 1.050–1.075), drug abuse (OR = 1.340; 95% CI = 1.283–1.400), fluid and electrolyte disorders (OR = 1.902; 95% CI = 1.886–1.918), psychoses (OR = 1.765; 95% CI = 1.725–1.806), renal failure (OR = 1.059; 95% CI = 1.048–1.070), and weight loss (OR = 1.690; 95% CI = 1.667–1.714). Results of multivariable logistic regression identifying risk factors for delirium in ischemic and hemorrhagic stroke are presented in S2–S4 Tables.

### Post-stroke complications and delirium

Patients with post-stroke delirium exhibited significantly higher rates of various medical complications, including dysphagia (16.9%), acute myocardial infarction (6.9%), pneumonia (13.0%), urinary tract infection (21.8%), deep vein thrombosis (2.5%), pulmonary embolism (1.3%), and sepsis (11.5%), compared to those without post-stroke delirium ($p<0.001$) (Table 5). Furthermore, multivariate logistic regression analysis indicated an association between post-stroke delirium and dysphagia (OR=1.315; 95% confidence interval [CI] = 1.301–1.329), acute myocardial infarction (OR=1.157; 95% CI = 1.139–1.176), pneumonia (OR = 1.434; 95% CI = 1.416–1.453), urinary tract infection (OR = 1.685; 95% CI = 1.668–1.702), deep vein thrombosis (OR = 1.270; 95% CI = 1.235–1.307), pulmonary embolism (OR = 1.132; 95% CI = 1.090–1.176), and sepsis (OR = 2.364; 95% CI = 2.329–2.400) (Table 5). A forest plot of the major risk factors is shown in Fig 2.

## Discussion

PSD is a frequently observed neuropsychiatric complication among elderly individuals who have suffered a stroke. It is linked to adverse outcomes, including extended hospital stays, higher mortality rates, and delayed recovery. Factors such

**Table 2. Patient characteristics and outcomes of post-stroke delirium (2010–2019).**

| Characteristics | No delirium | delirium | p |
|---|---|---|---|
| Total (n = count) | 1,324,152 | 320,621 | |
| Total incidence (%) | 80.5 | 19.5 | |
| Age (median, years) | 78.0 (71.0-84.0) | 79.0 (72.0-86.0) | <0.001 |
| Age group (%) | | | |
| 65-79 | 56.80 | 51.50 | <0.001 |
| ≥80 | 43.20 | 43.50 | |
| Sex (%) | | | |
| Male | 46.90 | 45.40 | <0.001 |
| Female | 53.10 | 54.60 | |
| Race (%) | | | |
| White | 73.40 | 71.40 | <0.001 |
| Black | 10.40 | 11.70 | |
| Hispanic | 6.00 | 6.20 | |
| Asian or Pacific Islander | 2.50 | 2.70 | |
| Native American | 0.40 | 0.40 | |
| Other | 7.40 | 7.60 | |
| Number of Comorbidities (%) | | | |
| 0 | 2.40 | 1.30 | <0.001 |
| 1 | 11.90 | 7.10 | |
| 2 | 21.00 | 14.00 | |
| ≥3 | 64.60 | 77.60 | |
| LOS (median, d) | 4 (2-6) | 5 (3-10) | <0.001 |
| TOTCHG (median, $) | 35,787 (20,633−67,170) | 44,863 (23,224−95,960) | <0.001 |
| Type of insurance (%) | | | |
| Medicare | 88.90 | 89.60 | <0.001 |
| Medicaid | 1.40 | 1.50 | |
| Private insurance | 7.70 | 6.90 | |
| Self-pay | 0.60 | 0.60 | |
| No charge | 0.00 | 0.00 | |
| Other | 1.30 | 1.40 | |
| Bed size of hospital (%) | | | |
| Small | 15.20 | 14.20 | <0.001 |
| Medium | 27.30 | 26.50 | |
| Large | 57.50 | 59.30 | |
| Elective admission (%) | 87.10 | 91.60 | <0.001 |
| Type of hospital (teaching %) | 61.10 | 61.30 | 0.041 |
| Location of hospital (urban, %) | 90.50 | 90.90 | <0.001 |
| Region of hospital (%) | | | |
| Northeast | 19.30 | 16.00 | <0.001 |
| Midwest or North Central | 22.80 | 23.10 | |
| South | 40.40 | 41.80 | |
| West | 17.50 | 19.10 | |
| Died (%) | 7.00 | 12.60 | <0.001 |

LOS: Length of stay, TOTCHE: Total charge.

**Table 3. Risk factors associated with post-stroke delirium.**

| Variable | Multivariate Logistic Regression | | |
|---|---|---|---|
| | OR | 95% CI | *p* |
| Age ≥ 80 years old | 1.237 | 1.227-1.247 | <0.001 |
| Female | 1.006 | 0.998-1.014 | 0.160 |
| Race | | | |
| White | Ref | —— | —— |
| Black | 1.113 | 1.099-1.127 | <0.001 |
| Hispanic | 1.005 | 0.989-1.022 | 0.531 |
| Asian or Pacific Islander | 1.060 | 1.034-1.087 | <0.001 |
| Native American | 1.022 | 0.959-1.089 | 0.497 |
| Other | 1.094 | 1.077-1.110 | <0.001 |
| Number of Comorbidities | | | |
| 0 | Ref | —— | —— |
| 1 | 1.040 | 1.004-1.077 | 0.028 |
| 2 | 1.149 | 1.111-1.188 | <0.001 |
| ≥3 | 2.049 | 1.984-2.116 | <0.001 |
| Type of insurance | | | |
| Medicare | Ref | —— | —— |
| Medicaid | 1.037 | 1.003-1.071 | 0.031 |
| Private insurance | 0.954 | 0.940-0.969 | <0.001 |
| Self-pay | 0.951 | 0.904-1.001 | 0.052 |
| No charge | 0.880 | 0.719-1.076 | 0.212 |
| Other | 1.100 | 1.064-1.137 | <0.001 |
| Bed size of hospital | | | |
| Small | Ref | —— | —— |
| Medium | 1.020 | 1.007-1.033 | 0.003 |
| Large | 1.083 | 1.070-1.095 | <0.001 |
| Elective admission | 0.681 | 0.672-0.690 | <0.001 |
| Teaching hospital | 0.991 | 0.982-1.000 | 0.041 |
| Urban hospital | 1.023 | 1.007-1.038 | 0.003 |
| Region of hospital | | | |
| Northeast | Ref | —— | —— |
| Midwest or North Central | 1.200 | 1.185-1.215 | <0.001 |
| South | 1.250 | 1.235-1.264 | <0.001 |
| West | 1.290 | 1.272-1.307 | <0.001 |

AIDS: Acquired immunodeficiency syndrome, OR: Odds ratio, CI: Confidence interval.

as advanced age, existing comorbid conditions, and stroke-related complications significantly elevate the risk of PSD, highlighting the importance of tailored preventive and management approaches. A large cohort of elderly stroke patients was examined to determine the incidence and risk factors associated with PSD. Our study determined an overall incidence of post-stroke delirium of 19.5% (Table 2), lower than the reported prevalence in the literature (25%) [2]. Besides, Fleischmann et al. reported a 39.0% prevalence among stroke patients over 60 years old [15]. Discrepancies between studies may arise from limitations in prior research, such as small sample sizes and an overrepresentation of elderly patients. These factors may have led to an inflated estimation of delirium prevalence following acute stroke. Moreover,

**Table 4. Relationship between pre-stroke comorbidities and post-stroke delirium.**

| Comorbidities | Univariate Analysis | | | Multivariate Logistic Regression | | |
|---|---|---|---|---|---|---|
| | No delirium | Delirium | p | OR | 95% CI | p |
| Pre-stroke comorbidities, n (%) | | | | | | |
| Acquired immune deficiency syndrome | 1,264 (0.1) | 308 (0.1) | 0.921 | 0.851 | 0.749-0.966 | 0.013 |
| Alcohol abuse | 32,064 (2.4) | 9,802 (3.10) | <0.001 | 1.126 | 1.099-1.153 | <0.001 |
| Deficiency anemia | 132,420 (10.0) | 42,062 (13.1) | <0.001 | 1.150 | 1.136-1.164 | <0.001 |
| Rheumatoid arthritis/collagen vascular diseases | 34,500 (2.60) | 8,817 (2.7) | <0.001 | 1.023 | 0.999-1.048 | 0.063 |
| Chronic blood loss anemia | 8,876 (0.7) | 2,435 (0.8) | <0.001 | 0.956 | 0.913-1.001 | 0.056 |
| Congestive heart failure | 227,917 (17.2) | 69,967 (21.8) | <0.001 | 1.186 | 1.174-1.198 | <0.001 |
| Coagulopathy | 74,568 (5.6) | 28,720 (9.0) | <0.001 | 1.333 | 1.314-1.353 | <0.001 |
| Depression | 132,876 (10.0) | 39,377 (12.3) | <0.001 | 1.236 | 1.221-1.251 | <0.001 |
| Diabetes with chronic complications | 169,315 (12.8) | 47,067 (14.7) | <0.001 | 1.062 | 1.050-1.075 | <0.001 |
| Drug abuse | 7,988 (0.6) | 2,965 (0.9) | <0.001 | 1.340 | 1.283-1.400 | <0.001 |
| Hypothyroidism | 225,079 (17.0) | 56,172 (17.5) | <0.001 | 1.012 | 1.002-1.023 | 0.019 |
| Liver disease | 21,184 (1.6) | 6,757 (2.1) | <0.001 | 1.036 | 1.007-1.066 | 0.015 |
| Fluid and electrolyte disorders | 306,051 (23.1) | 124,579 (38.9) | <0.001 | 1.902 | 1.886-1.918 | <0.001 |
| Psychoses | 25,367 (1.9) | 11,164 (3.5) | <0.001 | 1.765 | 1.725-1.806 | <0.001 |
| Pulmonary circulation disorders | 52,698 (4.0) | 15,224 (4.7) | <0.001 | 1.008 | 0.989-1.027 | 0.439 |
| Renal failure | 257,135 (19.4) | 72,822 (22.7) | <0.001 | 1.059 | 1.048-1.070 | <0.001 |
| Peptic ulcer disease excluding bleeding | 4,398 (0.3) | 1,370 (0.4) | <0.001 | 1.092 | 1.026-1.162 | 0.006 |
| Weight loss | 70,018 (5.3) | 34,322 (10.7) | <0.001 | 1.690 | 1.667-1.714 | <0.001 |

OR: Odds ratio, CI: Confidence interval.

**Table 5. Relationship between post-stroke delirium and post-stroke complications.**

| Complications | Univariate Analysis | | | Multivariate Logistic Regression | | |
|---|---|---|---|---|---|---|
| | No delirium | Delirium | p | OR | 95% CI | p |
| Medical complications, n (%) | | | | | | |
| Dysphagia | 163,445 (12.3) | 54,025 (16.9) | <0.001 | 1.315 | 1.301-1.329 | <0.001 |
| Acute myocardial infarction | 67,116 (5.1) | 22,014 (6.9) | <0.001 | 1.157 | 1.139-1.176 | <0.001 |
| Pneumonia | 95,257 (7.2) | 41,750 (13.0) | <0.001 | 1.434 | 1.416-1.453 | <0.001 |
| Urinary tract infection | 167,390 (12.6) | 69,966 (21.8) | <0.001 | 1.685 | 1.668-1.702 | <0.001 |
| Deep vein thrombosis | 19,677 (1.5) | 7,911 (2.5) | <0.001 | 1.270 | 1.235-1.307 | <0.001 |
| Pulmonary embolism | 11,394 (0.9) | 4,167 (1.3) | <0.001 | 1.132 | 1.090-1.176 | <0.001 |
| Sepsis | 52,829 (4.0) | 36,894 (11.5) | <0.001 | 2.364 | 2.329-2.400 | <0.001 |

OR: Odds ratio, CI: Confidence interval.

variations in diagnostic criteria and the accuracy of delirium identification across researchers and institutions may influence the reliability of reported outcomes. The findings also indicate that the incidence of delirium following hemorrhagic stroke is higher than that following ischemic stroke and the overall prevalence of post-stroke delirium, consistent with previous research [9]. Mechanisms plausibly include hematoma-associated release of blood-breakdown products and reactive oxygen species leading to robust neuroinflammation and increased exposure to ICU-related precipitating factors (e.g., mechanical ventilation, sedatives) [2,16,17]. These interacting processes likely account for the observed increased delirium risk after hemorrhagic stroke. Importantly, because delirium in our analysis was identified solely through ICD-9

| Risk factors | OR | 95% CI | p | |
|---|---|---|---|---|
| Sepsis | 2.364 | 2.329-2.400 | <0.001 | |
| Number of Comorbidities ≥ 3 | 2.049 | 1.984-2.116 | <0.001 | |
| Fluid and electrolyte disorders | 1.902 | 1.886-1.918 | <0.001 | |
| Psychoses | 1.765 | 1.725-1.806 | <0.001 | |
| Weight loss | 1.690 | 1.667-1.714 | <0.001 | |
| Urinary tract infection | 1.685 | 1.668-1.702 | <0.001 | |
| Pneumonia | 1.434 | 1.416-1.453 | <0.001 | |
| Drug abuse | 1.340 | 1.283-1.400 | <0.001 | |
| Coagulopathy | 1.333 | 1.314-1.353 | <0.001 | |
| Dysphagia | 1.315 | 1.301-1.329 | <0.001 | |
| West hospital | 1.290 | 1.272-1.307 | <0.001 | |
| Deep vein thrombosis | 1.270 | 1.235-1.307 | <0.001 | |
| South hospital | 1.250 | 1.235-1.264 | <0.001 | |
| Age ≥80 years old | 1.237 | 1.227-1.247 | <0.001 | |
| Depression | 1.236 | 1.221-1.251 | <0.001 | |
| Midwest or North Central hospital | 1.200 | 1.185-1.215 | <0.001 | |

**Fig 2. Forest plot of major risk factors for post-stroke delirium.**

and ICD-10 diagnostic codes, misclassification and undercoding in administrative data likely led to underascertainment of true delirium cases and an underestimation of the observed incidence [18].

Our study highlights the significant impact of post-stroke delirium on clinical outcomes. Patients with delirium experienced extended hospital stays (5 days vs. 4 days, $p<0.001$), increased hospitalization expenses ($44,863 vs. $35,787, $p<0.001$), and elevated in-hospital mortality rates (12.6% vs. 7.0%, $p<0.001$) (Table 2). These results aligned with prior research indicating that delirium is a prevalent and severe complication in stroke patients, contributing to cognitive and functional decline, prolonged hospitalization, and increased mortality risk [2,19,20].

Our analysis revealed racial disparities, with Black and Asian/Pacific Islander patients demonstrating an increased risk of developing delirium (Table 3). These findings aligned with previous studies emphasizing racial disparities in health outcomes [21,22]. Although these associations reached statistical significance, the magnitude of the effects was small. Moreover, because our analysis is based on a large administrative dataset (the National Inpatient Sample, NIS), the substantial sample size can make modest between-group differences statistically significant without implying substantial clinical impact [23]. Possible explanations for the observed disparities include socioeconomic and healthcare access differences, cultural factors influencing symptom recognition and help-seeking, and unmeasured clinical or social confounders; in addition, we cannot rule out the possibility that racial differences in diagnostic coding or in the identification of delirium within administrative data may have influenced the observed associations. Clinically, while the modest effect sizes temper claims about a large individual-level impact, these findings nonetheless identify patient groups who may warrant closer clinical monitoring and tailored delirium-prevention interventions [24]. Further work using richer clinical data is warranted to clarify causal pathways and to estimate absolute risk increases in more clinically granular cohorts.

Multiple studies on post-stroke delirium have highlighted the critical role of early detection and intervention in managing risk factors to enhance patient outcomes [5,9]. Consequently, understanding these risk factors is essential for developing targeted interventions. Our findings are largely consistent with previous research and provide further insights into the predictors of post-stroke delirium. Logistic regression analysis indicated that individuals aged ≥80 years had a significantly higher likelihood of developing delirium (Table 3). This finding aligned with prior research emphasizing advanced age as a critical risk factor for delirium [15]. The mechanisms underlying this association are complex and may involve age-related physiological changes in brain function, as well as increased susceptibility to metabolic disturbances during stroke recovery [6,25]. Furthermore, the likelihood of delirium was elevated in patients presenting with ≥3 comorbidities (Table 3). This finding aligned with previous studies indicating that the presence of multiple comorbid conditions correlates with a poorer

stroke prognosis [26,27]. Among these comorbidities, alcohol abuse and deficiency anemia were associated with delirium (Table 4), indicating that these conditions may exacerbate the neurocognitive decline in stroke patients [28–30]. In addition to patient-related physiological factors, hospital bed capacity was independently associated with post-stroke delirium in our logistic regression analysis. Notably, hospitals with greater bed sizes (detailed specifications for bed sizes are available at the following URL: https://hcup-us.ahrq.gov/db/nation/nis/nisdbdocumentation.jsp) exhibited the highest incidence of delirium, which may reflect ascertainment bias arising from heightened diagnostic vigilance that enhances delirium detection [31]. Specifically, such hospitals are typically large tertiary care centers equipped with advanced medical technologies and staffed by highly specialized personnel, thereby contributing to elevated detection rates [31,32].

As expected, fluid and electrolyte disorders exhibited the strongest association with post-stroke delirium, with an odds ratio (Table 4), corroborating prior evidence that fluid and electrolyte disturbances can affect neuronal function, impair the nervous system, and significantly increase the risk of delirium [29,33,34]. Moreover, a prior history of neuropsychiatric disorders, such as psychoses (OR = 1.765) and depression, was associated with an elevated risk of delirium (Table 4). This finding underscores the contribution of pre-existing brain pathology to delirium development [29,35,36]. Systemic comorbidities were also identified as independent risk factors. Renal failure and congestive heart failure have been previously linked to delirium (Table 4), likely due to their impact on cerebral perfusion and metabolic stability [16,37,38]. Notably, our findings indicate that drug abuse is an important risk factor for post-stroke delirium. Research indicates that the abuse of sedatives, in particular, may impair the function of extensive neural networks by suppressing central nervous system activity and disrupting GABAergic and adrenergic neurotransmission, thereby inducing delirium [16,39]. These findings emphasize the necessity of optimizing comorbidity management to mitigate delirium risk and enhance the overall prognosis in elderly stroke patients. Although several risk factors, such as advanced age, prior neurological disorders, and psychiatric comorbidities, are non-modifiable, their identification remains clinically significant. Recognizing high-risk patients allows for targeted delirium prevention strategies, such as enhanced monitoring, early mobilization, and cognitive interventions. In our subgroup analyses, pre-existing hypothyroidism showed statistically significant associations with post-stroke delirium in both ischemic and hemorrhagic stroke patients. Several plausible biological mechanisms could explain why pre-existing hypothyroidism might increase the risk of delirium after ischemic stroke. Hypothyroid states promote suboptimal cerebral circulation, which increase the likelihood and severity of ischemic cerebral injury; greater ischemic burden and impaired cerebral perfusion reserve may in turn raise the brain's vulnerability to acute cognitive disturbances such as delirium [40]. In addition, thyroid hormone deficiency is associated with pro-inflammatory and oxidative stress pathways and can impair blood–brain barrier integrity and neuronal resilience, mechanisms implicated in delirium pathogenesis [41]. Finally, hypothyroidism is linked to chronic cognitive impairment and neuropsychiatric symptoms that reduce cognitive reserve and thereby predispose patients to delirium when exposed to the acute physiological stress of ischemic stroke [42]. Therefore, close monitoring of thyroid function may serve as an effective strategy for preventing post-stroke delirium in elderly patients.

Notably, private insurance and elective admission were both associated with a reduced adjusted risk of post-stroke delirium (Table 3). These associations are more likely to reflect broader socioeconomic disparities and baseline health differences. Research indicates that lack of private insurance or other indicators of social disadvantage are independently associated with an increased risk of delirium, and that social determinants including healthcare accessibility, caregiver support, and community disadvantage play an important role in delirium susceptibility [43,44]. Among older adults, private insurance typically represents greater economic resources, improved access to primary and specialty care, and enhanced availability of social support and caregiver assistance, the latter facilitates medication adherence, follow-up management, and early recognition of clinical changes [45]. These advantages contribute to improved chronic disease control (e.g., heart failure, diabetes) and more timely management of acute precipitating factors (e.g., infection, electrolyte imbalance), collectively reducing the risk of delirium in frail stroke patients [16,46,47]. Elective admission also implies lower baseline disease severity and greater opportunities for pre-admission optimization, such as medication reconciliation, correction of metabolic abnormalities, treatment of concurrent infections, and patient/caregiver education [48,49]. Consequently,

compared to unplanned emergency admissions, elective admissions carry lower short-term physiological vulnerability to delirium [48]. Although both associations were statistically significant, the effect sizes were modest, and the large NIS sample may have amplified small between-group differences without clear clinical significance at the individual level [23]. Moreover, private insurance and elective admission were associated with lower odds of post-stroke delirium in the overall cohort and among ischemic stroke patients, but not in the hemorrhagic subgroup. Administrative coding (e.g., ICD) is known to have low sensitivity for detecting delirium and may differentially under capture cases across stroke types and care settings [18]. Thus, any protective effect of private insurance and elective admission may be detectable in less-fatal ischemic strokes but obscured in the clinically more severe hemorrhagic cohort [43,50]. Residual confounding effects from baseline stroke severity and other unmeasured factors may also contribute. These findings identify system-level factors (insurance status, admission pathway) that merit further investigation and, if confirmed in clinically detailed cohorts, could guide targeted strategies to reduce delirium incidence.

In subgroup analyses by stroke type, hospital characteristics showed divergent associations with post-stroke delirium. We observed that admission to a teaching hospital was associated with lower odds of post-stroke delirium in ischemic stroke patients, but this association was not significant in the overall cohort or the hemorrhagic subgroup. Teaching hospital and certified stroke centers typically deliver organized stroke-unit care, multidisciplinary pathways, and rapid reperfusion, and they more often implement routine delirium screening and prevention, factors that likely account for the reduced recorded delirium in the ischemic subgroup [51]. Additionally, we found that admission to an urban hospital was associated with delirium only among patients with hemorrhagic stroke. Urban centers often care for patients with a higher burden of comorbidity and adverse social determinants of health, both independently linked to delirium, which may partially explain the observed association [52]. In addition, referral and ascertainment biases, whereby severe hemorrhagic stroke cases are preferentially transferred to tertiary urban hospitals and undergo more intensive monitoring, could amplify the association in the hemorrhagic subgroup relative to ischemic stroke [52]. Thus, the higher delirium prevalence after hemorrhagic stroke seen in urban hospitals likely reflects a combination of greater baseline severity, ICU-related risks, and differential case ascertainment rather than a direct causal effect of hospital location.

The occurrence of delirium was significantly influenced by post-stroke complications. Patients experiencing post-stroke complications such as dysphagia, pneumonia, urinary tract infection, and sepsis exhibited a higher risk of delirium (Table 5). The literature has consistently shown that other complications in hospitalized stroke patients tend to induce or worsen delirium [2,15]. Our findings suggest that the management of these complications should be prioritized in the prevention of post-stroke delirium [19]. Additionally, evidence indicates that several post-stroke complications may causally contribute to delirium. A cohort study of critically ill patients with ischemic stroke showed that dysphagia increases delirium risk, partially mediated by reduced serum albumin levels [53]. Elderly patients with pneumonia are likewise more prone to delirium, with malnutrition and systemic inflammation acting as key intermediaries [54]. Experimental data further suggest that urinary tract infections can precipitate delirium via inflammatory mechanisms, while delirium itself may secondarily increase UTI risk through dehydration, urinary retention, or greater exposure to indwelling catheters, indicating a potential bidirectional relationship [55,56]. Sepsis, in particular, is a well-established trigger of acute brain dysfunction: sepsis-related systemic inflammation disrupts the blood–brain barrier and induces microcirculatory and mitochondrial dysfunction, neuroinflammation, and neurotransmitter imbalance, ultimately impairing attention and arousal networks and manifesting as delirium [57]. Collectively, these findings underscore the need for large, high-quality prospective studies to clarify causal pathways between post-stroke delirium and associated complications, thereby informing targeted prevention and management strategies.

In summary, demographic factors, socioeconomic status, comorbidity burden, and post-stroke complications converge to create a high-risk state for delirium after stroke. To translate these findings into practice, we recommend routine delirium-risk assessment on admission for all older stroke patients and stratified prevention for those at greatest risk — for example, patients aged ≥80 years, with ≥3 comorbidities, a history of drug abuse, fluid and electrolyte disorders or

psychoses, or post-stroke complications such as sepsis, urinary tract infection, pneumonia, or dysphagia [4,5,9]. High-risk patients should receive intensified surveillance of cognitive and psychological status, vigilance for infection, and proactive management of fluids, nutrition, and electrolytes, together with careful medication review to minimize exposure to deliriogenic drugs [5,58]. Finally, a multidisciplinary approach involving neurology, geriatrics, nursing, pharmacy, and rehabilitation is essential to implement targeted prevention and early-intervention strategies to reduce delirium incidence and improve outcomes [58,59].

## Limitations

While this research offers critical insights into the incidence and risk factors associated with post-stroke delirium, several inherent limitations must be acknowledged. The retrospective design of this study limits our capacity to infer causal relationships, while the reliance on administrative data may have led to underreporting or inaccuracies in the coding of variables such as delirium, thereby potentially underestimating the incidence of post-stroke delirium. Furthermore, the lack of standardized delirium screening protocols in this study, together with potential diagnostic alertness bias arising from heterogeneity in hospital settings (e.g., higher detection rates in tertiary care facilities), may contribute to discrepancies between reported post-stroke delirium incidence and its true prevalence. Meanwhile, the lack of data in the database regarding key risk factors for delirium, notably stroke severity, pre-stroke cognitive status, drug type and frequency of administration, may have limited the precision of our estimates. Moreover, because the NIS lacks information on hospital-level delirium screening tools, prevention practices, and patient-specific management strategies, delirium may be differentially detected, potentially biasing observed associations through misclassification and residual confounding. Additional prospective research is required to validate these results and investigate the mechanisms through which factors like age and comorbidities influence the development of post-stroke delirium.

## Conclusions

This large-scale study of elderly stroke patients (n = 1,644,773) found an overall post-stroke delirium (PSD) incidence of 19.5%, with 18.9% in ischemic stroke and 24.7% in hemorrhagic stroke. Several independent risk factors were identified. Advanced age (≥80 years), increased comorbidity burden (particularly ≥3 comorbidities, OR = 2.049), and specific disorders, such as fluid and electrolyte disorders (OR = 1.902) and psychosis (OR = 1.765), were strongly associated with the development of delirium. Patients with delirium exhibited a higher mortality rate (12.6% vs. 7.0%), longer hospitalization, and an increased incidence of complications, notably sepsis (OR = 2.364). These results highlight the importance of promptly identifying patients at elevated risk and implementing tailored preventive measures within stroke management protocols for the elderly population.

## Supporting information

**S1 File. ICD-9-CM and ICD-10-CM diagnosis codes.**
(PDF)

**S1 Table. Incidence of delirium following ischemic and hemorrhagic strokes.**
(DOCX)

**S2 Table. Multivariable logistic regression analysis of associations between sociodemographic factors and risk of delirium following ischemic and hemorrhagic stroke in elderly patients.**
(DOCX)

**S3 Table. Multivariable logistic regression analysis of associations between comorbidities and risk of delirium following ischemic and hemorrhagic stroke in elderly patients.**
(DOCX)

**S4 Table. Multivariable logistic regression analysis of associations between complications and risk of delirium following ischemic and hemorrhagic stroke in elderly patients.**
(DOCX)

**S5 Table. Diagnosis of multicollinearity among comorbidity and complication factors.**
(DOCX)

## Author contributions

**Conceptualization:** Ying Gao, Jingqin Wang.

**Data curation:** Yaoyang Huo, Hao Xie.

**Formal analysis:** Jianrong Zhang, Mei Chen, Yaoyang Huo, Yu'e Wu, Xilin Liu, Xiaohuan Li, Jingqin Wang, Fengling Yang, Gang Liu.

**Investigation:** Ying Gao, Jianrong Zhang, Mei Chen, Yaoyang Huo, Yu'e Wu, Xilin Liu, Xiaohuan Li, Jingqin Wang, Fengling Yang, Gang Liu, Hao Xie.

**Methodology:** Jianrong Zhang, Mei Chen, Yaoyang Huo, Yu'e Wu, Xilin Liu, Xiaohuan Li, Hao Xie.

**Project administration:** Ying Gao.

**Software:** Yaoyang Huo, Hao Xie.

**Supervision:** Ying Gao, Jingqin Wang, Fengling Yang, Gang Liu, Hao Xie.

**Validation:** Ying Gao, Jianrong Zhang, Mei Chen, Yaoyang Huo, Yu'e Wu, Xilin Liu, Xiaohuan Li.

**Visualization:** Jianrong Zhang, Mei Chen, Yaoyang Huo, Yu'e Wu, Xilin Liu, Xiaohuan Li.

**Writing – original draft:** Ying Gao, Jianrong Zhang, Mei Chen, Yaoyang Huo, Yu'e Wu, Xilin Liu, Xiaohuan Li, Jingqin Wang, Fengling Yang, Gang Liu.

**Writing – review & editing:** Ying Gao, Jianrong Zhang, Mei Chen, Yaoyang Huo, Yu'e Wu, Xilin Liu, Xiaohuan Li, Gang Liu, Hao Xie.

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
