## [Decision Letter · Decision Letter 0]

27 Nov 2025

Dear Dr. Gao,

Thank you for submitting your manuscript to PLOS ONE. After careful consideration, we feel that it has merit but does not fully meet PLOS ONE’s publication criteria as it currently stands. Therefore, we invite you to submit a revised version of the manuscript that addresses the points raised during the review process.

We look forward to receiving your revised manuscript.

Kind regards,

Kamalakar Surineni, MD, MPH

Guest Editor

PLOS ONE

Journal Requirements:

3. We notice that your supplementary tables are included in the manuscript file. Please remove them and upload them with the file type 'Supporting Information'. Please ensure that each Supporting Information file has a legend listed in the manuscript after the references list.

Reviewers' comments:

Reviewer's Responses to Questions

**Comments to the Author**

1. Is the manuscript technically sound, and do the data support the conclusions?

Reviewer #1: Yes

Reviewer #2: Yes

2. Has the statistical analysis been performed appropriately and rigorously?

Reviewer #1: Yes

Reviewer #2: Yes

3. Have the authors made all data underlying the findings in their manuscript fully available?

Reviewer #1: Yes

Reviewer #2: Yes

4. Is the manuscript presented in an intelligible fashion and written in standard English?

Reviewer #1: Yes

Reviewer #2: Yes

Reviewer #1: Strengths:

1. Large, nationally representative dataset — enhances statistical power and generalizability.

2. Timely clinical relevance — post-stroke delirium is an under-recognized yet impactful complication influencing outcomes and costs.

3. Robust statistical analysis — appropriate use of multivariate logistic regression with adjustment for confounders.

4. Clear organization and presentation — tables and figures convey findings effectively.

Weaknesses and Recommendations

1. Methodological Clarifications:

Definition of Delirium:

The study identifies delirium based on ICD-9/10 codes, which are prone to undercoding. Please acknowledge this limitation explicitly and discuss how misclassification bias could affect the reported incidence.

Stroke Subtypes:

Clarify which ICD codes were used to identify ischemic and hemorrhagic stroke. Consider a sensitivity analysis by stroke type, if feasible.

Multicollinearity:

Explain how potential overlap between comorbidities (e.g., sepsis, metabolic disorders, infection) was addressed in regression modeling.

2.Interpretation and Discussion

Some predictors (e.g., race, elective admission) have small odds ratios but are statistically significant. Discuss clinical vs. statistical significance and interpret findings within the context of real-world relevance.

Expand discussion on protective factors (e.g., private insurance, elective admission) — likely reflecting socioeconomic and health status differences.

Highlight that delirium identification may be influenced by diagnostic vigilance bias — patients in tertiary centers may have higher detection rates.

3.Presentation and Organization

Abstract: Include the most impactful predictors with adjusted odds ratios to strengthen clinical takeaways.

Tables/Figures: Consider combining redundant figures and adding a forest plot summarizing major risk factors.

Language and Grammar: Minor copyediting suggested for precision and readability. Example:

“Patients in the delirium group exhibited a higher proportion of patients aged 80 years and above” → “The delirium group had a greater proportion of patients aged ≥80 years.”

4.Literature and References

Some references are duplicated (e.g., 8 and 13). Please revise for consistency.

Incorporate recent literature (2023–2024) on delirium prediction and prevention after stroke.

Reviewer #2: Review Summary:

This article is well-written, with concise statements and accessible language. The study presents valuable insights but also has notable limitations.

Strengths:

Inclusion of a broad population across multiple geographic and demographic groups enhances generalizability. Both modifiable and non-modifiable risk factors are considered. The data and findings are clearly laid out, aiding reader comprehension.

Weaknesses:

1. Being a retrospective Design, it limits diagnostic accuracy of delirium and the assessment of influencing risk factors.

2. Severity and subtype of stroke are not specified, which could significantly affect outcomes.

3. Bed size is unclear, and teaching status is mentioned without explaining its impact.

4. It’s unclear whether delirium led to complications (e.g., dysphagia, pneumonia, UTI) or vice versa.

5. Absence of information on delirium screening and prevention methods may affect outcome interpretation.

6. Baseline Cognitive Status: Not reported, which is a critical factor in delirium risk.

7. Medication Data: Missing, despite its relevance to delirium onset and progression.

Recommendations:

1. Include ICD Codes and this will address potential undercoding or miscoding of delirium.

2. Clarify Stroke Subtypes and Cognitive Baseline: These are essential for accurate risk stratification.

3. Discuss Prevention Implications: Especially for high-risk groups, to inform hospital delirium prevention programs.

Conclusion:

The article offers meaningful contributions to the field. With the above gaps addressed, it would be suitable for publication.

**Do you want your identity to be public for this peer review?** For information about this choice, including consent withdrawal, please see our Privacy Policy

Reviewer #1: **Yes:** VENKATA VIJAYA K DALAI

Reviewer #2: **Yes:** Harish Pulluru

---

## [Author Response · Author response to Decision Letter 1]

29 Dec 2025

Dear Editors and Reviewers,

We appreciate the opportunity to revise our manuscript entitled " Incidence and Risk Factors for Post-Stroke Delirium in the Elderly: A National Inpatient Sample (NIS) Analysis. " We are grateful to your thoughtful and constructive comments, which have significantly improved the quality and clarity of our work and will help guide our future research. Below we provide detailed, point-by-point responses to each your comment. All changes in the manuscript are highlighted in red. Content requiring deletion is indicated with a red strikethrough. In addition to addressing the specific points raised, we have undertaken a comprehensive revision of the entire manuscript to enhance clarity and presentation. In this response letter, your comments are presented in italics, and our corresponding changes and additions to the manuscript are highlighted in red text. The page numbers and line numbers cited in the reply correspond to the locations in the manuscript bearing revision marks. We have tried our best to make all the revisions clear, and we hope that the revised manuscript meets the requirements for publication.

Reviewer #1:

Comment 1: Methodological Clarifications: Definition of Delirium: The study identifies delirium based on ICD-9/10 codes, which are prone to undercoding. Please acknowledge this limitation explicitly and discuss how misclassification bias could affect the reported incidence.

Response 1: Thank you for your insightful feedback. First, we acknowledge that the use of ICD-based administrative data is subject to underreporting bias and that identifying delirium solely through diagnostic codes may underestimate its true prevalence. To mitigate this limitation as far as possible, we adopted a comprehensive ICD-9/10 code set for delirium that was systematically compiled from previously published, peer-reviewed NIS and other administrative-database studies [1,2] to enhance the comparability and validity of our findings.

The revised discussion now reads:

Importantly, because delirium in our analysis was identified solely through ICD-9 and ICD-10 diagnostic codes, misclassification and undercoding in administrative data likely led to underascertainment of true delirium cases and an underestimation of the observed incidence [3]. (Page 15, line 232-234)

References Cited in the Response:

[1] Kumar M, Patil S, Godoy LDC, Kuo CL, Swede H, Kuchel GA, et al. Demand Ischemia as a Predictor of Mortality in Older Patients With Delirium. Front Cardiovasc Med. 2022;9:917252. https://doi.org/10.3389/fcvm.2022.917252 PMID: 35734279

[2] Taha A, Xu H, Ahmed R, Karim A, Meunier J, Paul A, et al. Medical and economic burden of delirium on hospitalization outcomes of acute respiratory failure: A retrospective national cohort. Medicine (Baltimore). 2023;102(2):e32652. https://doi.org/10.1097/MD.0000000000032652 PMID: 36637939

[3] Sheehan KA, Shin S, Hall E, Mak DYF, Lapointe-Shaw L, Tang T, et al. Characterizing medical patients with delirium: a cohort study comparing ICD-10 codes and a validated chart review method. PLoS One. 2024;19(5):e0302888. https://doi.org/10.1371/journal.pone.0302888 PMID: 38739670

Comment 2: Stroke Subtypes: Clarify which ICD codes were used to identify ischemic and hemorrhagic stroke. Consider a sensitivity analysis by stroke type, if feasible.

Response 2: Thank you for your thoughtful comments. We have revised the Discussion and Results (S1 Table in S2 File) to incorporate stroke-subtype sensitivity analyses (stratified multivariable logistic regressions using the same covariate set as the primary models; ICD case definitions are provided in Supplementary Material S1 File). Key points now discussed and linked to Supplementary S2–S4 Tables in S2 File are: (1) delirium incidence is higher after hemorrhagic than ischemic stroke — likely driven by hematoma-related blood-breakdown products, neuroinflammation, and greater exposure to ICU-level precipitating factors; (2) private insurance and elective admission show a protective association in the overall cohort and the ischemic subgroup but not in hemorrhagic patients, which may reflect differential case severity, monitoring, or coding sensitivity; (3) admission to a teaching hospital was associated with lower recorded delirium in ischemic stroke (not in hemorrhagic), plausibly due to organized stroke-unit care and routine delirium prevention/screening; (4) the association between urban hospitals and delirium appeared only for hemorrhagic stroke and is likely influenced by referral/case-mix and ascertainment bias rather than a direct causal effect; and (5) pre-existing hypothyroidism was associated with increased delirium risk in both subgroups, for which we propose biological mechanisms (cerebral perfusion, inflammation/oxidative stress, and reduced cognitive reserve) and recommend closer thyroid monitoring. We also explicitly acknowledge residual confounding, differential ICD coding sensitivity and selection biases as limitations that temper inference.

The revised objective now reads:(See S1-S4 Tables in S2 File.)

The revised results now reads:

PSD occurred in 18.9% of ischemic strokes and 24.7% of hemorrhagic strokes (S1 Table in S2 File).

(Page 8, line 142-143)

The revised discussion now reads:

The findings also indicate that the incidence of delirium following hemorrhagic stroke is higher than that following ischemic stroke and the overall prevalence of post-stroke delirium, consistent with previous research [1]. Mechanisms plausibly include hematoma-associated release of blood-breakdown products and reactive oxygen species leading to robust neuroinflammation and increased exposure to ICU-related precipitating factors (e.g., mechanical ventilation, sedatives) [2-4].

(Page 14-15, line 226-230)

In our subgroup analyses, pre-existing hypothyroidism showed statistically significant associations with post-stroke delirium in both ischemic and hemorrhagic stroke patients. Several plausible biological mechanisms could explain why pre-existing hypothyroidism might increase the risk of delirium after ischemic stroke. Hypothyroid states promote suboptimal cerebral circulation, which increase the likelihood and severity of ischemic cerebral injury; greater ischemic burden and impaired cerebral perfusion reserve may in turn raise the brain’s vulnerability to acute cognitive disturbances such as delirium [5]. In addition, thyroid hormone deficiency is associated with pro-inflammatory and oxidative stress pathways and can impair blood–brain barrier integrity and neuronal resilience, mechanisms implicated in delirium pathogenesis [6]. Finally, hypothyroidism is linked to chronic cognitive impairment and neuropsychiatric symptoms that reduce cognitive reserve and thereby predispose patients to delirium when exposed to the acute physiological stress of ischemic stroke [7]. Therefore, close monitoring of thyroid function may serve as an effective strategy for preventing post-stroke delirium in elderly patients.

(Page 18, line 295-307)

Moreover, private insurance and elective admission were associated with lower odds of post-stroke delirium in the overall cohort and among ischemic stroke patients, but not in the hemorrhagic subgroup. Administrative coding (e.g., ICD) is known to have low sensitivity for detecting delirium and may differentially undercapture cases across stroke types and care settings [8]. Thus, any protective effect of private insurance and elective admission may be detectable in less-fatal ischemic strokes but obscured in the clinically more severe hemorrhagic cohort [9,10]. Residual confounding effects from baseline stroke severity and other unmeasured factors may also contribute.

(Page 19-20, line 326-332)

In subgroup analyses by stroke type, hospital characteristics showed divergent associations with post-stroke delirium. We observed that admission to a teaching hospital was associated with lower odds of post-stroke delirium in ischemic stroke patients, but this association was not significant in the overall cohort or the hemorrhagic subgroup. Teaching hospital and certified stroke centers typically deliver organized stroke-unit care, multidisciplinary pathways, and rapid reperfusion, and they more often implement routine delirium screening and prevention, factors that likely account for the reduced recorded delirium in the ischemic subgroup [11]. Additionally, we found that admission to an urban hospital was associated with delirium only among patients with hemorrhagic stroke. Urban centers often care for patients with a higher burden of comorbidity and adverse social determinants of health, both independently linked to delirium, which may partially explain the observed association [12]. In addition, referral and ascertainment biases, whereby severe hemorrhagic stroke cases are preferentially transferred to tertiary urban hospitals and undergo more intensive monitoring, could amplify the association in the hemorrhagic subgroup relative to ischemic stroke [12]. Thus, the higher delirium prevalence after hemorrhagic stroke seen in urban hospitals likely reflects a combination of greater baseline severity, ICU-related risks, and differential case ascertainment rather than a direct causal effect of hospital location.

(Page 20, line 336-351)

References Cited in the Response:

[1] Zhang G-B, Li H-Y, Yu W-J, Ying Y-Z, Zheng D, Zhang X-K, et al. Occurrence and risk factors for post-stroke delirium: a systematic review and m-eta-analysis. Asian J Psychiatry. 2024;99:104132. https://doi.org/10.1016/j.ajp.2024.104132 PMID: 38981150

[2] Zhang G-B, Lv J-M, Yu W-J, Li H-Y, Wu L, Zhang S-L, et al. The associations of post-stroke delirium with outcomes: A systematic review and me-ta-analysis. BMC Med. 2024;22(1):470. https://doi.org/10.1186/s12916-024-03689-1 PMID: 39407191

[3] Fan Y-Y, Luo R-Y, Wang M-T, Yuan C-Y, Sun Y-Y, Jing J-Y. Mechanisms underlying delirium in patients with critical illness. Front Aging Neurosci. 2024;16. https://doi.org/10.3389/fnagi.2024.1446523 PMID: 39391586

[4] Krishnan K, Campos PB, Nguyen TN, Tan CW, Chan SL, Appleton JP, et al. Cerebral edema in intracerebral hemorrhage: pathogenesis, natural history, and potential treatments from translation to clinical trials. Front Stroke. 2023;2. https://doi.org/10.3389/fstro.2023.1256664

[5] Tian Y, Shi XQ, Shui JW, Liu XY, Bu Y, Liu Y, et al. Exploring the cau-sal factor effects of hypothyroidism on ischemic stroke: A two-sample men-delian randomization study. Front Neurol. 2024;15. https://doi.org/10.3389/fneur.2024.1322472 PMID: 38361639

[6] Zúñiga D, Balasubramanian S, Mehmood KT, Al-Baldawi S, Zúñiga Salazar G. Hypothyroidism and cardiovascular disease: a review. Cureus. 2024;16(1):e52512. https://doi.org/10.7759/cureus.52512 PMID: 38370998

[7] Sheng X, Gao J, Chen K, Zhu X, Wang Y. Hyperthyroidism, hypothyroidis-m, thyroid stimulating hormone, and dementia risk: Results from the NHA-NES 2011–2012 and mendelian randomization analysis. Front Aging Neurosci. 2024;16. https://doi.org/10.3389/fnagi.2024.1456525 PMID: 39507203

[8] Sheehan KA, Shin S, Hall E, Mak DYF, Lapointe-Shaw L, Tang T, et al. Characterizing medical patients with delirium: a cohort study comparing ICD-10 codes and a validated chart review method. PLoS One. 2024;19(5):e0302888. https://doi.org/10.1371/journal.pone.0302888 PMID: 38739670

[9] Nguyen MTH, Sakamoto Y, Maeda T, Woodward M, Anderson CS, Catiwa J, et al. Influence of socioeconomic status on functional outcomes after stroke: a systematic review and meta-analysis. J Am Heart Assoc. 2024;13(9):e033078. https://doi.org/10.1161/JAHA.123.033078 PMID: 38639361

[10] Dagli C, Huang Z, Lin C. Intracerebral hemorrhage hospitalizations and outcomes: Comparisons between institutional and national data. Intern Emerg Med. 2025;20(5):1455–1462. https://doi.org/10.1007/s11739-025-03977-5 PMID: 40437327

[11] Shen Y-C, Kim AS, Hsia RY. Treatments and patient outcomes following s-troke center expansion. JAMA Netw Open. 2024;7(11):e2444683. https://doi.org/10.1001/jamanetworkopen.2024.44683 PMID: 39535793

[12] Loggini A, Hornik J, Schwertman A, Hornik A. Rural-urban disparities in acute stroke treatments and outcomes: A propensity score-matched analysis of a nationwide sample. Cerebrovasc Dis. 2025;1-9. https://doi.org/10.1159/000546950 PMID: 40562010

Comment 3: Multicollinearity: Explain how potential overlap between comorbidities (e.g., sepsis, metabolic disorders, infection) was addressed in regression modeling.

Response 3: Thank you for raising this important point. In the submitted revision we have already performed VIF-based collinearity diagnostics and report that all independent variables returned VIF values below 5. The results of diagnosing multicollinearity among comorbidity and complication factors are detailed in S5 Table of S2 File.

The revised method now reads:

Univariate and multivariate logistic regression analyses were performed to identify independent risk factors for post-stroke delirium; before fitting the multivariate models, multicollinearity diagnostics were conducted separately for the two predictor categories (comorbidities and complications) [1].

(Page 7, line 130-133)

The revised result now reads:

Collinearity was assessed using variance inflation factors; all independent variables returned VIF values below 5, indicating that multicollinearity was not a concern (S5 Table in S2 File) [1].

(Page 10, line 172-173)

References Cited in the Response:

[1] Marcoulides KM, Raykov T. Evaluation of variance inflation factors in regr-ession models using latent variable modeling methods. Educ Psychol Meas. 2019;79(5):874–882. https://doi.org/10.1177/0013164418817803 PMID: 31488917

Comment 4: Interpretation and Discussion: Some predictors (e.g., race, elective admission) have small odds ratios but are statistically significant. Discuss clinical vs. statistical significance and interpret findings within the context of real-world relevance.

Response 4:

Thank you for this important point. We fully agree that statistical significance does not necessarily imply clinical importance, particularly for analyses of large administrative datasets such as the NIS. In response to your recommendation we have revised the Discussion to explicitly distinguish statistical from clinical significance for the protective factors (private insurance and elective admission) as well as for race, and to clarify plausible mechanisms and limitations.

The revised discussion now reads:

Although these associations reached statistical significance, the magnitude of the effects was small. Moreover, because our analysis is based on a large administrative dataset (the National Inpatient Sample, NIS), the substantial sample size can make modest between-group differences statistically significant without implying substantial clinical impact [1]. Possible explanations for the observed disparities include socioeconomic and healthcare access differences, cultural factors influencing symptom recognition and help-seeking, and unmeasured clinical or social confounders; in addition, we cannot rule out the possibility that racial differences in diagnostic coding or in the identification of delirium within administrative data may have influenced the observed associations. Clinically, while the modest effect sizes temper claims about a large individual-level impact, these findings nonetheless identify patient groups who may warrant closer clinical monitoring and tailored delirium-prevention interventions [2]. Further work using richer clinical data is warranted to clarify causal pathways and to estimate absolute risk increases in more clinically granular cohorts.

(Page 15-16, line 243-255)

Elective admission also implies lower baseline disease severity and greater opportunities for pre-admission optimization, such as medication reconciliation, correction of metabolic abnormalities, treatment of concurrent infections, and patient/caregiver education [3,4]. Consequently, compared to unplanned emergency admissions, elective admissions carry lower short-term physiological vulnerability to delirium [3]. Although both associations were statistically significant, the effect sizes were modest, and the large NIS sample may have amplified small between-group differences without clear cl

---

## [Editor Report · Decision Letter 1]

13 Jan 2026

Incidence and risk factors for post-stroke delirium in the elderly: a national inpatient sample (NIS) analysis

PONE-D-25-43559R1

Dear Dr. Gao

We’re pleased to inform you that your manuscript has been judged scientifically suitable for publication and will be formally accepted for publication once it meets all outstanding technical requirements.

Kind regards,

Kamalakar Surineni, MD, MPH

Guest Editor

PLOS One

Additional Editor Comments (optional):

Thank you so much for revising the manuscript based on the reviewer feedback. I can see that you addressed all required components and met the criteria for acceptance.
---

## [Editor Report · Acceptance letter]

PONE-D-25-43559R1

PLOS One

Dear Dr. Gao,

I'm pleased to inform you that your manuscript has been deemed suitable for publication in PLOS One. Congratulations! Your manuscript is now being handed over to our production team.

Kind regards,

on behalf of

Dr. Kamalakar Surineni

Guest Editor

PLOS One